# The Impact of K-1 Kickboxing Tournament Fights on Growth Hormone, IGF-1, and Insulin Levels: Preliminary Findings from a Pilot Study in Elite Athletes

**DOI:** 10.3390/jcm14207203

**Published:** 2025-10-13

**Authors:** Łukasz Rydzik, Ismail Ilbak, Serkan Düz, Tadeusz Ambroży, Tomasz Pałka, Marta Kopańska, Marta Niewczas, Anna Kurkiewicz-Piotrowska, Ibrahim Ouergui, Monika Bigosińska, Wojciech Wąsacz

**Affiliations:** 1Faculty of Physical Education and Sport, Institute of Sports Sciences, University of Physical Culture, 31-571 Kraków, Poland; tadeusz.ambrozy@awf.krakow.pl (T.A.); wojciech.wasacz@awf.krakow.pl (W.W.); 2Institute of Health Sciences, İnönü University, 44050 Malatya, Türkiye; isma_ilbak@hotmail.com; 3Faculty of Sports Sciences, Department of Coaching Education, İnönü University, 44050 Malatya, Türkiye; serkan.duz@inonu.edu.tr; 4Department of Physiology and Biochemistry, Faculty of Physical Education and Sport, University of Physical Culture, 31-571 Krakow, Poland; tomasz.palka@awf.krakow.pl; 5Department of Medical Psychology, Faculty of Medicine, University of Rzeszów, al. Tadeusza Rejtana 16C, 35-959 Rzeszów, Poland; mkopanska@ur.edu.pl; 6Institute of Physical Culture Studies, College of Medical Sciences, University of Rzeszow, 35-959 Rzeszow, Poland; 7Institute for Basic Sciences, Faculty of Physiotherapy, University of Physical Culture, 31-571 Kraków, Poland; anna.piotrowska@awf.krakow.pl; 8High Institute of Sport and Physical Education of Kef, University of Jendouba, Kef 7100, Tunisia; ouergui.brahim@yahoo.fr; 9Faculty of Physical Education and Security Sciences, Academy of Applied Sciences in Nowy Sącz, 33-300 Nowy Sącz, Poland; mbigosinska@ans-ns.edu.pl

**Keywords:** combat sports, sports confrontation, growth hormone, insulin-like growth factor 1, insulin

## Abstract

**Background:** Evidence on acute endocrine responses to K-1 kickboxing is limited. This pilot study assessed pre-to-post changes in GH, IGF-1 and insulin after a single K-1 bout and explored relationships with training experience (TE), final heart rate (HRFINAL) and perceived exertion (RPE). **Methods:** Elite male K-1 athletes (n = 10) completed an interclub, referee-supervised three-round bout (3 × 2 min). Venous blood was sampled pre-fight and +2 min. GH, IGF-1 and insulin were assayed (ELISA). Paired *t*-tests and Pearson’s r quantified changes and associations. **Results:** GH rose from 1.20 ± 2.05 to 11.27 ± 8.82 ng/mL (*p* = 0.007; d = 1.85), and insulin from 5.95 ± 1.56 to 12.95 ± 5.95 µU/mL (*p* = 0.002; d = 1.86); IGF-1 showed no change (200.90 ± 55.89 to 203.00 ± 54.10 ng/mL; *p* = 0.497). ΔGH and Δinsulin correlated positively with HRFINAL (rp = 0.89 and 0.88, both *p* < 0.001) and RPE (rp = 0.70 and 0.68; *p* = 0.024 and 0.031), and negatively with TE (rp = −0.72 and −0.68; *p* = 0.019 and 0.031). **Conclusions:** A single K-1 fight elicited large acute increases in GH and insulin but not IGF-1 at +2 min. HRFINAL and RPE tracked response magnitude, while more experienced athletes showed smaller deltas. Findings are preliminary and limited by a small sample, elite-only cohort, in an interclub setting, with immediate sampling and absence of a control group.

## 1. Introduction

Combat sports, especially striking-oriented ones like kickboxing, have a large global fan base [1,2,3]. Kickboxing competitions are performed under different styles based on the variety of techniques used and the degree of contact force allowed [4,5], with K-1 being one of the most popular categories [6,7]. In this context, K-1 kickboxing, which allows both hand and leg strikes, including techniques from classical boxing such as punches, as well as spinning and jumping punches, in addition to various types of kicks, and knee strikes as performed in Muay Thai [8,9]. It is also one of the most popular kickboxing sports which is characterised by rapid, high-paced, and intense physical contact [10]. Kickboxing comprises several rulesets that differ in permitted techniques and clinch rules. In “classical” formats (e.g., full-contact or low-kick), prolonged clinching is prohibited and knee strikes are generally not allowed; elbows and throws are also banned, and exchanges are frequently broken to reset distance. By contrast, the K-1 ruleset allows brief clinch actions to deliver an immediate knee strike (while still banning elbows and prolonged holds) and emphasises continuous stand-up striking with punches and a wide range of kicks. These differences typically increase the density of striking exchanges and perceived intensity in K-1 compared with classical styles, which helps explain why K-1 is considered more specific to sustained, high-intensity stand-up combat and may provoke distinct acute endocrine responses [9,11].

K-1 is a short-duration, repeated high-intensity striking format (typically 3 × 2-min rounds with 1-min rests) that combines alactic bursts (explosive entries, single strikes) with a predominant anaerobic–glycolytic contribution during sustained exchanges. Compared with “classical” kickboxing (more frequent resets) and boxing (punch-only), K-1 permits brief clinch actions to deliver knees and emphasises continuous stand-up striking, thereby recruiting more muscle mass and increasing exchange density. This profile is expected to produce higher lactate accumulation and catecholamine drive, i.e., a stronger acute endocrine stimulus than styles with fewer continuous exchanges [9,10,12,13].

Kickboxing K-1 matches are classified among the most intense sports that push kickboxers’ physical and physiological limits [9,10]. It is well known that high-intensity physical activity strongly disrupts the body’s homeostasis, making the possible changes in hormonal responses a subject of many studies [14]. Research investigating the interactions between exercise and hormones has frequently focused on factors such as growth hormone (GH), insulin-like growth factor-1 (IGF-1) and insulin [15].

GH is a crucial hormone that affects numerous systems in the body [16]. It influences carbohydrate, lipid, amino acid and protein metabolism, promoting the growth and development of nearly all tissues capable of growing [17]. GH elicits these effects through IGF-1, induced in target tissues [18]. Additionally, IGF-1 promotes glycogen synthesis [19], tendon collagen synthesis [19], and may indirectly support GH-induced lipolysis [20]. The anabolic hormone insulin is regulated by blood glucose and amino acid levels, playing a key role in skeletal muscle growth [15].

While the exercise-induced GH response is well known [21], it is assumed that IGF-1 release in the liver is stimulated by GH increase [22,23]. However, many studies have emphasised that increased IGF-1 production does not always follow the same pattern as GH changes [24,25,26,27]. Therefore, it is suggested that their releases may be independent [28], and the effect of exercise on circulating IGF-1 levels is not yet definitively known [29]. Hence, further research in this area is important.

Previous research in striking combat sports has focused mainly on classical kickboxing and boxing. These studies analyzed hormonal responses, including changes in GH, IGF-1, and insulin levels after training or competition [13]. On this basis, K-1 competition would be expected to elicit a robust acute GH rise—as consistently observed after high-intensity striking efforts—while IGF-1 may show minimal or delayed change within very early sampling windows, and insulin may increase transiently in line with post-bout glucose handling and glycogen restoration. We therefore hypothesised large pre-to-post increases in GH and insulin and a variable, potentially null, immediate IGF-1 response in K-1 bouts [13,21,30]. Prior work in striking sports indicates a robust but heterogeneous endocrine profile: GH typically rises after high-intensity efforts and competitions [28,31]. IGF-1 responses are mixed—some studies report increases [32], others decreases, and several show no acute change with very brief, intense efforts. Insulin tends to [33,34] show transient post-exercise elevations consistent with glucose handling and glycogen restoration, with post-bout glucose rises also observed in simulated kickboxing [35,36]. These discrepancies, together with the specific demands of the K-1 ruleset, justify the present pilot study and the chosen sampling window. Although important insights into hormonal reactions in combat conditions have been obtained, there is a lack of studies addressing the K-1 formula, which is characterised by a distinct specificity (knee strikes, limited clinch, high intensity). There is a need for a more detailed exploration of this area, particularly in the context of real tournament fights, as previous reports have not fully accounted for the unique characteristics of this discipline. From a practical standpoint, acute post-bout surges in GH and insulin indicate high glycolytic demand and physiological stress, which should inform periodisation and recovery planning. Coaches can space K-1–specific high-intensity sessions, schedule lighter technical work or active recovery within the next 24–48 h and ensure adequate carbohydrate availability to support glycogen restoration. Simple field markers—final heart rate and CR-10 RPE—can be used to titrate weekly load (e.g., bouts with near-max HR and RPE ≥ 8 treated as high-stress exposures). Because IGF-1 often shows delayed changes, periodic (e.g., weekly) tracking may help detect emerging fatigue; conversely, persistently blunted GH/insulin responses alongside high perceived effort could flag non-functional overreaching and prompt a short deload [13,21,29]. We focused on GH, IGF-1 and insulin because they provide complementary information about acute stress and recovery processes relevant to striking performance. GH is highly sensitive to exercise intensity and metabolic acidosis and rises within minutes, indexing acute internal load [21]. Circulating IGF-1 reflects GH-dependent hepatic signalling and tissue anabolism (e.g., glycogen and tendon collagen synthesis) but often changes on a slower timescale, informing post-exercise adaptation rather than immediate stress [18,19,31]. Insulin integrates carbohydrate availability and skeletal–muscle glucose uptake, underpinning rapid glycogen resynthesis after intense efforts [37,38]. Together, this triad links bout intensity to short-term fuel management and longer-term anabolic recovery, which is directly actionable for periodisation and return-to-training decisions in K-1 [20,29].

The study had a clearly defined hierarchy of aims. The primary aim was to assess the acute pre-to-post changes in GH, IGF-1 and insulin after a single K-1 fight. The specific aims were (i) to analyse the relationships of ΔGH, ΔIGF-1 and Δinsulin with TE, HRFINAL and CR-10 RPE, and (ii) to provide effect-size estimates to guide sample-size planning for future research. In view of the small sample (n = 10), the study is framed as a pilot with preliminary findings. Based on previous scientific reports, the main hypothesis was that a real K-1 fight would elicit a significant increase in growth hormone and insulin levels in the studied athletes. We also proposed a functional hypothesis, assuming that the rating of perceived exertion (RPE) protocol may serve as a simple and useful tool for assessing the intensity of the hormonal response after the fight.

## 2. Materials and Methods

The project obtained approval from the Ethics Committee at the District Medical Chamber in Krakow, under the reference number No. 226/KBL/OIL/2023. In accordance with the Helsinki Declaration requirements, the participants were informed about the research objectives, methods, potential side effects, and the option to withdraw from the study at any time without providing a reason. The participants provided written consent to participate.

### 2.1. Study Design

A cross-sectional study was conducted in elite K-1 kickboxers who regularly compete at national and international levels. To determine the impact of a K-1 bout on serum GH, IGF-1, and insulin, we organised an internal interclub tournament. Bouts followed WAKO K-1 rules under a licensed referee and were matched by official weight categories [12]. Venous blood was drawn immediately before and 2 min after each fight and processed as described in Section 2.4. Immediately post-bout we recorded final heart rate (HRFINAL) and collected training experience (TE) and CR-10 rating of perceived exertion (RPE) to examine their relationships with the hormonal response.

To minimise variance from non-exercise factors, all bouts and sample collections were scheduled within a fixed late-afternoon window (≈16:00–19:00) to reduce diurnal effects. Participants were instructed to (i) maintain their habitual diet for 48 h prior to testing; (ii) arrive post-absorptive (no caloric intake in the final 2–3 h before the bout); (iii) abstain from alcohol and caffeine for 24 h, nicotine for 2 h, and strenuous exercise for 24 h; and (iv) avoid acute medications or supplements known to affect glucose/insulin or GH/IGF-1 signalling. Athletes reported typical sleep duration from the preceding night; current illness or injury constituted an exclusion criterion.

### 2.2. Participants

Sample size and study framing. An a priori calculation was conducted in G*Power version 3.1.9.7 (Heinrich Heine University Düsseldorf, Düsseldorf, Germany), with α = 0.05 and 1 − β = 0.80. In the initial submission we illustrated a “large” standardised effect (d = 0.90). We now explicitly acknowledge that such an effect size is optimistic in a hormonal context with high inter-individual variability. Accordingly, this investigation is framed as a pilot study designed to generate variance and effect-size estimates (with 95% confidence intervals) to inform adequately powered confirmatory trials. Given typical variability in endocrine outcomes, moderate effects would require substantially larger samples; thus, all results should be interpreted as preliminary.

The study included a purposefully selected group of 10 elite athletes who train and actively compete in kickboxing under K-1 rules (male participants). A homogeneous sampling technique was applied to minimise internal variability within the sample (similar experience, training specificity, and fitness level), which enhances measurement precision. Accordingly, the inclusion criteria for the study were: at least five years of training experience, absence of current musculoskeletal injuries, a positive medical recommendation, and active participation in competitions. Exclusion criteria included the opposite of the aforementioned variables, a history of severe injuries, any use of doping substances, being on a reduction diet and supplementation in the period prior to (at least 30 days) and during the study, as well as the athlete’s refusal to participate in the study. In addition, a medical interview was conducted, which confirmed the absence of carbohydrate metabolism pathologies (e.g., diabetes, insulin resistance) among the recruited participants.

All participants competed in international, national and local elite tournaments. Some of them achieved significant sporting results, including medals in European, national, and local prestigious kickboxing competitions. The study was conducted during the athletes’ preparatory period. Information regarding chronological age and training and competition experience was obtained through a diagnostic survey conducted in the form of direct interviews with the athletes and coaching staff.

Body height was measured using a stadiometer (SECA, Hamburg, Germany) with an accuracy of 0.01 m, while body mass was assessed using a certified electronic scale, TANITA TBF-538 (TANITA, Tokyo, Japan), with an accuracy of 0.1 kg. Anthropometric measurements of the participants were conducted according to the techniques and standards recommended by the International Society for the Advancement of Kinanthropometry (ISAK) [39].

The average body mass of the participants was 73.56 ± 7.96 kg, height 174.16 ± 5.63 cm, BMI 24.21 ± 3.02, and age 28.08 ± 4.72 years. Their training experience averaged 7.40 ± 2.01 years (including: n = 2 with 5 years of training experience; n = 2 with 6 years; n = 2 with 7 years; n = 1 with 8 years; and n = 3 with 10 years of training experience) and included systematic training with 4 to 6 sessions per week, depending on the implemented mesocycle.

### 2.3. K-1 Tournament Fights

An interclub internal tournament was organised. Five matches were conducted strictly according to the rules of the World Association Kickboxing Organization (WAKO) [12]. Each athlete participated in a single three-round bout (each round lasting 2 min) with a 1-min break between rounds. The matchups were deliberately arranged based on weight category. The confrontations took place in a 6 × 6 m ring, supervised by an experienced, licensed referee. During the bouts, all participants wore 10-ounce boxing gloves, protective helmets, discipline-specific foot and shin guards, as well as mouth and groin protectors. All matches were conducted for the full regulatory duration, and no knockouts were recorded.

Before the bouts, the participants underwent a standard 20-min warm up designed to prepare the body for physical exertion. The warm up included jogging, static and dynamic movements of the arms, forearms and wrists, trunk movements in all planes, as well as exercises for the abdomen and lower limbs (hip, knee and ankle joints). Additionally, K-1-specific exercises were performed, such as shadow boxing and boxing footwork drills. This interclub format was selected specifically to enable immediate pre- and post-bout blood sampling under competition-like conditions while complying with safety and organisational constraints that preclude venipuncture at official international events.

### 2.4. Blood Sample Collection and Analysis

An experienced and qualified specialist collected venous blood samples from the participants using the venipuncture method at two different time points: immediately before (resting level) and two minutes after the fight (post-confrontation effect). Venous blood samples were drawn under sterile conditions from the antecubital vein using a 21-gauge butterfly needle (BD Vacutainer®, Becton, Dickinson and Company, Franklin Lakes, NJ, USA) and collected into 5 mL EDTA tubes (BD Vacutainer^®^, Becton, Dickinson and Company, Franklin Lakes, NJ, USA). Participants remained seated throughout the blood collection. Participants were ensured to be seated during the blood collection process.

The collected blood samples were centrifuged at 4000 rpm for 10 min within 30 min of collection to separate the plasma. Plasma samples were stored at −40 °C until the time of analysis. The blood samples were placed in biochemistry tubes and maintained under a cold chain. These samples were then transported to the laboratory for analysis. Levels of GH, IGF-1 and insulin were measured using the ELISA (Enzyme-Linked Immunosorbent Assay) method with commercial kits (R&D Systems, Minneapolis, MN, USA). ELISA tests were conducted according to the manufacturer’s protocols. Absorbance values were measured at 450 nm using a microplate reader (BioTek Instruments, Winooski, VT, USA), and hormone concentrations were calculated using standard curves. Rationale for the +2-min window. We selected an immediate post-fight window (~+2 min) to capture rapid endocrine responses to high-intensity effort (particularly GH and insulin) while minimising interference with ringside safety and event logistics. However, circulating IGF-1 can exhibit delayed kinetics relative to GH/insulin; therefore, peak IGF-1 responses may not be captured within this early window [29,40,41]

### 2.5. Measurements of Objective and Subjective Indicators of Exercise Intensity

Heart rate, expressed in beats per minute (bpm), was measured in each participant immediately after the fight (HR_FINAL_). Monitoring was carried out using a Polar 610S sport tester (Polar Electro Oy, Kempele, Finland).

Subjective perception of exertion (RPE) was assessed using the CR-10 scale [42]. One minute after the fight, the athletes responded to a question concerning their perceived exertion on a scale from 0 to 10, where 0 indicated no exertion, 5 represented moderate effort with the ability to continue comfortably, and 10 corresponded to maximal exertion requiring immediate cessation and rest. All participants had been instructed and were familiar with the scale prior to testing. RPE assessment protocol. Perceived exertion was obtained using the CR-10 scale following standard instructions [34]. To capture end-bout exertion without disrupting ringside procedures, athletes reported CR-10 approximately 1 min after the fight. Immediate post-exercise CR-10 is widely used for high-intensity tasks; we note that a session-RPE approach (≈30-min post-session) is also common but was not feasible in this ringside setting. Future studies may include both immediate and delayed (session-RPE) assessments for triangulation [13,42].

### 2.6. Statistical Analysis

The data were analysed in SPSS 25.0 (IBM, Armonk, NY, USA). Descriptive statistics are reported as mean ± SD. Distribution shape was inspected with skewness and kurtosis (±2 SD). Pre–post differences were tested with paired-samples *t*-tests (two-tailed, α = 0.05). Alongside *p*-values, we report standardised effect sizes and 95% confidence intervals (CI) to convey estimate precision in this small pilot sample [43]. For group differences we computed Cohen’s d (or Hedges’ g when bias-corrected values are reported), and for associations Pearson’s r. Effect sizes were interpreted using discipline-relevant thresholds for physiotherapy/exercise sciences Archives of Physical Medicine and Rehabilitation, 2025 (APMR 2025): for r, small = 0.3, medium = 0.5, large = 0.6; for d/g, small = 0.1, medium = 0.4, large = 0.8 [44]. Correlation significance was set at *p* < 0.05.

## 3. Results

Table 1 presents a comparative summary of the diagnosed hormone levels before and after the confrontation among the examined kickboxing/K-1 athletes.

It was found that GH and insulin increased significantly after the combats (*p* = 0.007 and 0.002, respectively), indicating a large effect (d = 1.85 and 1.86). A substantial difference was observed in the GH levels’ increase after the fight sessions. The mean GH concentration after the fight increased more than ninefold compared to the pre-fight values. A similar significant trend was reported for the increase in insulin levels after competitions, in which mean concentration rose by 118% compared to the pre-intervention level. No statistically significant increase was recorded for IGF-1 concentration.

Within the studied group, the mean training experience (TE) was 7.40 ± 2.01 years, the final heart rate (HR_FINAL_) immediately after the fight was 183.30 ± 6.36 bpm, and the rating of perceived exertion (RPE) was 8.20 ± 0.79. Pre–post deltas with 95% CI were as follows: GH Δ = +10.07 ng/mL (95% CI 3.56 to 16.59), *p* = 0.007, d = 1.85 (large); IGF-1 Δ = +2.10 ng/mL (95% CI −4.61 to 8.81), *p* = 0.497, d = 0.04; insulin Δ = +7.00 µU/mL (95% CI 3.46 to 10.54), *p* = 0.002, d = 1.86 (large).

Table 2 presents the analysis of relationships between post-fight hormonal levels and objective and subjective indicators of exertion.

Significant strong negative correlations were found between TE and the increase in GH and insulin. A similar trend was observed for the relationships of RPE and HR_FINAL_ with the increase in GH and insulin, although in this case the associations were positive. Moreover, the correlations with HR_FINAL_ showed a very strong magnitude.

Pearson correlations with 95% CI: ΔGH with TE, rp = −0.72 (95% CI −0.93 to −0.17), *p* = 0.019; with HRFINAL, rp = 0.89 (0.59 to 0.97), *p* < 0.001; with RPE, rp = 0.70 (0.13 to 0.92), *p* = 0.024. ΔIGF-1 showed non-significant associations: TE, rp = −0.42 (−0.83 to 0.29); HRFINAL, rp = 0.44 (−0.26 to 0.84); RPE, rp = 0.41 (−0.30 to 0.83). ΔInsulin correlated with TE, rp = −0.68 (−0.92 to −0.09), *p* = 0.031; HRFINAL, rp = 0.88 (0.56 to 0.97), *p* < 0.001; and RPE, rp = 0.68 (0.09 to 0.92), *p* = 0.031.

## 4. Discussion

The present study examined the acute effects of an interclub K-1 tournament on athletes’ GH, IGF-1 and insulin levels. It was observed that there was a statistically significant increase in the athletes’ GH and insulin levels; however, IGF-1 levels did not statistically change across combats. Another key finding of this study is the demonstrated pattern of relationships between the activity of the analysed hormones and the objective and subjective exercise variables. Athletes with shorter training experience, along with higher HR_FINAL_ and RPE values after the fight, exhibited higher levels of GH and insulin.

While it was expected that GH and insulin release levels will increase after kickboxing combat effort due to the intense nature of K-1 style; it is unlikely that a short-term acute physical effort will not show a significant difference in the level of IGF-1, which presents a complex regulatory process. Although it has been reported that serum IGF-1 levels generally increase during aerobic exercise and decrease during anaerobic efforts, our findings are not consistent with these studies [45]. Similarly, serum GH and IGF-1 levels decreased after exercise in elite kickboxers compared to healthy controls [46]. As well, previous studies conducted on boxers and kickboxers reported that serum GH and IGF-1 levels were significantly lower in kickboxers compared to healthy controls [28].

A body of evidence showed that both aerobic and anaerobic exercises induce an acute increase in GH levels [47]. Additionally, Luger et al. [48] reported increased plasma GH concentrations induced by acute exercise in three groups of healthy male volunteers. Although it has been reported that both ski and football trainings increase serum GH levels, IGF-1 did not change significantly. The absence of an acute IGF-1 change despite a large GH surge is physiologically plausible given sampling and exercise modality. Circulating (hepatic) IGF-1 often shows delayed or blunted alterations in the first minutes after high-intensity efforts, with several studies reporting no change at ~0–5 min despite later shifts, and a dependence on exercise duration and modality [29,49]. The predominantly anaerobic, high-intensity profile of K-1 bouts [9] may preferentially elicit rapid GH elevations while leaving total IGF-1 unchanged within a +2-min window, particularly as bioavailability is further modulated by IGF-binding proteins (e.g., acute IGFBP-1 fluctuations) [50]. In contrast, increases are more likely with longer or resistance-type stimuli and at later sampling times [24,51], whereas prolonged endurance loads can even reduce IGF-1 [52]. Accordingly, our null IGF-1 finding should be interpreted as timing- and format-specific rather than as evidence of no endocrine involvement; future K-1 studies should include additional post-bout time points (e.g., +15, +30, +60 min and beyond) and, where feasible, indices of free/bioactive IGF-1 to capture the full kinetics.

Taken together, our null change in total IGF-1 at +2 min supports the position that very early sampling after short, predominantly anaerobic efforts often yields no immediate IGF-1 alteration, despite a large GH surge. The inconsistencies across studies are largely explained by methodological factors—sampling timing (early minutes vs. later), exercise modality/duration (glycolytic vs. oxidative/resistance volume), energy and catecholamine milieu, and IGF-binding protein dynamics—rather than fundamental disagreement. In this K-1 context, the +2-min null is therefore expected, and future work should include later time points and bioactive/free IGF-1 to resolve kinetics.

Acute IGF-1 responses to exercise are heterogeneous. Decreases after acute exercise have been reported (e.g., Barnard et al. [53]), whereas increases in GH-deficient subjects [54] indicate that post-exercise IGF-1 elevations are not simply driven by contemporaneous exercise-induced GH surges [55]. Consistently, measurements of total IGF-1 in serum/plasma have yielded inconsistent acute results [29]. Moreover, intensive training may negatively affect adipose tissue hormones [56], altering the endocrine milieu and contributing to between-study variability. Thus, apparent discrepancies are likely due to methodological and biological factors—sampling timing (early minutes vs. later), exercise modality and duration (glycolytic vs. oxidative/resistance), and individual differences. In the present K-1 context (predominantly anaerobic profile with very early sampling at +2 min), the absence of an immediate change in total IGF-1 is timing- and format-consistent.

Separately, given repeated head impacts in striking sports, regular monitoring of pituitary function should be considered. Recurrent blows can injure the pituitary stalk and lead to traumatic brain-injury-mediated hypopituitarism, with documented cases in kickboxers [57]. Such injury can dysregulate the hypothalamic–pituitary–adrenal (HPA) axis and GH/IGF-1 pathways, potentially blunting or altering acute endocrine reactivity to combat stressors [13,14]. Neurophysiological alterations observed after K-1 fights further support the biological plausibility of head-impact-related endocrine vulnerability in this population [6]. Therefore, beyond short-term bout responses, a practical priority is longitudinal pituitary screening (especially after concussive/subconcussive exposure) and clinical vigilance for symptoms compatible with HPA- or GH-axis dysfunction (e.g., persistent fatigue, mood/sleep disturbance, reduced performance). Similarly, Ouergui et al. also reported a significant increase in plasma glucose level after a simulated full-contact kickboxing match, supporting our study’s findings [56].

The nature of the associations observed in this study suggests that training experience may modulate the hormonal response to a fight. GH and insulin showed strong positive correlations with HR_FINAL_ and RPE, indicating that greater intensity and physiological stress were accompanied by a stronger hormonal response. In contrast, training experience demonstrated a strong negative correlation, showing that longer training backgrounds were associated with smaller increases in GH and insulin after the intervention. The absence of such correlations between IGF-1 and both objective and subjective indicators of exertion suggests that IGF-1 is less sensitive to short-term stimuli. The smaller ΔGH and Δinsulin in more experienced athletes can be interpreted in at least two, non-exclusive ways. First, with training age athletes often exhibit greater metabolic efficiency and a dampened endocrine response to a given external/internal load—consistent with adaptations in sympathoadrenal drive, insulin sensitivity and substrate utilisation [21,22,36]. Under this view, a smaller perturbation reflects efficiency rather than “weaker” responsiveness. Second, chronic high loads can transiently blunt GH/IGF-1 axis reactivity (non-functional overreaching) and alter HPA-axis signalling, particularly in striking sports with repeated high-stress exposures [14,55]. Disentangling efficiency from suppression will require stratified designs (novice vs. elite), longitudinal tracking across mesocycles, and additional markers (e.g., cortisol, catecholamines, IGFBP-1/free IGF-1), as well as analyses that adjust for HR and RPE to test whether training experience predicts hormonal deltas beyond acute intensity.

Athletes with longer training backgrounds exhibited smaller increases in GH and insulin, which may reflect a higher level of metabolic adaptation. A smaller rise in GH/insulin may not indicate a weaker response, but rather a more efficient homeostatic mechanism [45]. In contrast, athletes with shorter training experience may display a more pronounced hormonal reaction, potentially indicating an overload of the system. This is consistent with previous findings showing that, among young, trained boxers, longer training experience was associated with lower hormonal variability [58]. The results also revealed a significant positive relationship between hormonal responses and exercise intensity assessed both objectively (HR) and subjectively (RPE), which aligns with other studies in this area [59,60]. Athletes experiencing greater physiological stress (higher perceived exertion and heart rate) demonstrated greater increases in GH/insulin activity. This may serve as a valuable diagnostic tool for monitoring an athlete’s condition without the need for invasive testing, and for adjusting training loads and recovery strategies. These findings also confirm our practical hypothesis regarding the usefulness of the RPE tool.

### Limitations of the Study

First, blood sampling was limited to two time points (pre-fight and +2 min post-fight), which prioritised immediate responses but may not capture the full time course of hormonal dynamics. In particular, circulating IGF-1 often shows delayed or blunted changes in the first minutes after high-intensity efforts, with some studies reporting no acute change at ~0–5 min despite later alterations, and a dependence on exercise duration and modality. Accordingly, future work should include additional post-bout time points (e.g., +15, +30, +60 min and subsequent hours) and/or area-under-the-curve analyses to better characterise peak timing and recovery kinetics. This study is a pilot with a very small sample (n = 10) of elite K-1 athletes. Although we performed an a priori calculation, the originally assumed large effect size (d = 0.90) is likely optimistic for hormonal outcomes with high inter-individual variability. Consequently, the present findings should be considered preliminary and primarily useful for planning adequately powered future studies. Fourth, this pilot included only elite male K-1 athletes and no control or comparison cohort, which constrains external validity and precludes determining whether the observed endocrine responses are specific to this population.

## 5. Conclusions

In this pilot sample of elite male K-1 kickboxers (n = 10), a single bout elicited significant pre-to-post increases in GH and insulin, whereas IGF-1 did not change at +2 min.GH and insulin changes tracked positively with HRFINAL and CR-10 RPE and inversely with training experience.These findings suggest that simple field markers (HRFINAL, RPE) can help flag high-stress exposures and inform immediate recovery planning without invasive testing.Generalisability is limited by the small sample, elite-only cohort, interclub setting, immediate sampling window, and absence of a control group; results should be interpreted as preliminary.Future studies should include larger and stratified cohorts, additional post-bout time points (e.g., +15, +30, +60 min and beyond), and complementary markers (e.g., free/bioactive IGF-1) to characterise kinetics and test specificity.

### Practical Implications

The results of this study may offer practical value for coaches and athletes in personalising training loads and recovery. The acute hormonal response after a K-1 fight can serve as an indicator of physiological load. Less-experienced athletes may show larger GH and insulin increases, suggesting greater metabolic stress and the need for longer recovery, whereas more-experienced athletes may be better adapted and able to tolerate higher loads. Simple field indicators—final heart rate (HRFINAL) and CR-10 rating of perceived exertion (RPE)—correlate with hormone levels and can be used to monitor training load without blood testing. An athlete presenting high HRFINAL and high RPE may require more recovery time and a temporary reduction in intensity.

For implementation, collect CR-10 RPE within ~10 min post-bout and use HRFINAL to flag acute endocrine stress. Treat RPE ≥ 8 and/or HRFINAL ≥ ~90% HRmax as high-stress exposures; cap these at ≤2 per week, separate by 24–48 h, and follow with carbohydrate repletion and lighter technical/active-recovery work. Keep weekly load progression (e.g., sRPE) to ≤10–15%. Red flags for a short deload include persistently high RPE (≥8) with rising fatigue, or high RPE alongside blunted HR responses relative to baseline.

## Figures and Tables

**Table 1 jcm-14-07203-t001:** Statistical characteristics and comparative analysis (pre- and post-confrontation) of hormonal variables in K-1 kickboxing athletes (n = 10).

Variables	Measurement	n	$\tilde{x}$	sd	t	df	*p*	d_C_
**GH**	pre	10	1.199	2.051	−3.496	9	0.007 *	1.85
	post	10	11.273	8.818				
**IGF-1**	pre	10	200.900	55.887	−0.708	9	0.497	0.04
	post	10	203.000	54.100				
**Insulin**	pre	10	5.950	1.560	−4.475	9	0.002 *	1.86
	post	10	12.950	5.947				

$\tilde{x}$—mean; sd—standard deviation; GH—growth hormone; IGF-1—insulin-like growth factor-1; t—paired *t*-test; df—degrees of freedom; d_C_—standardized effect size. Effect-size interpretation follows physiotherapy/exercise-science thresholds: small = 0.1, medium = 0.4, large = 0.8. * *p*, 0.05.

**Table 2 jcm-14-07203-t002:** Correlation coefficients between hormonal variables and objective and subjective indicators of physical exertion.

Variables	TE		HR_FINAL_		RPE	
	r_p_	*p*	r_p_	*p*	r_p_	*p*
**GH**	−0.72	0.019 *	0.89	<0.001 **	0.70	0.024 *
**IGF-1**	−0.42	0.227	0.44	0.203	0.41	0.239
**Insulin**	−0.68	0.031 *	0.88	<0.001 **	0.68	0.031 *

**GH**—growth hormone, **IGF-1**—insulin-like growth factor 1, **TE**—training experience, **HR_FINAL_**—heart rate final, **RPE**—rated perceived exertion, **r_p_**—value of the correlation coefficient Pearson, ***p***—level of significance, * statistically significant values (*p* < 0.05), ** statistically significant values (*p* < 0.001).

## Data Availability

The data presented in this study are available upon request from the corresponding author.

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
