# Peer review of "The Impact of K-1 Kickboxing Tournament Fights on Growth Hormone, IGF-1, and Insulin Levels: Preliminary Findings from a Pilot Study in Elite Athletes"

_jcm, 2025, doi:10.3390/jcm14207203_

Round 1

Reviewer 1 Report

Comments and Suggestions for Authors

After reviewing the manuscript, I have the following recommendations. The key issue is the small sample size, which in my opinion should be reflected in the title and objectives. The paper should be framed more clearly as preliminary findings or a pilot study.

  1. In the header “J. Clin. Med. 2024, 13, x FOR PEER REVIEW” the year is incorrect – it should be 2025.
  2. L58 – “[1-5]” – to support the statement, a maximum of three references is sufficient, not five.
  3. In the introduction, various formulas (classical kickboxing, K-1) are mentioned, but there is no clear comparison of their key differences (e.g. clinch rules, techniques prohibited in classical style, intensity of exchanges). Readers outside combat sports may struggle to understand why K-1 is considered “more specific”. Please remember that the paper will be read not only by specialists in the field.
  4. In the literature review section, it is noted that previous studies analysed GH, IGF-1 and insulin in kickboxing and boxing, but no concrete results or discrepancies are provided. Examples should be given (e.g. which studies confirmed an IGF-1 increase, which did not) to better justify the research gap.
  5. The introduction announces the importance of the results for athletes and coaches, but does not explain how potential hormonal changes might translate into training practice (e.g. periodisation, recovery, prevention of overtraining). A short paragraph linking hormonal responses with practical applications would strengthen the paper.
  6. In “The aim of this study” section, two aims are listed descriptively but not clearly separated. It is recommended to distinguish the main aim (e.g. to assess acute changes in GH, IGF-1 and insulin after a K-1 fight) from specific aims (e.g. to analyse relationships with training experience, heart rate, and RPE). This would improve clarity and scientific rigour.

Methods and Materials
7. Blood samples were taken only before and 2 minutes after the fight. Such a short interval may not capture the full dynamics of hormonal changes, particularly IGF-1, which often responds with a delay. This limitation should be explicitly acknowledged.
8. The sample size (n = 10) is very small – although power analysis was performed, the high individual variability in hormonal studies increases the risk of error. Moreover, the assumed effect size of 0.90 seems unrealistic in a hormonal context. I recommend reframing the study as a pilot or preliminary report and noting the potential error in the sample size calculation.
9. A strong aspect of the paper is the well-described study group.
10. Interclub matches may differ in intensity and psychological stress from official international competitions. This could influence hormonal responses and weaken ecological validity. The choice of match format should be justified.
11. No mention is made of controlling confounding variables (diet, time of day, psychological stress, sleep quality), which may significantly influence hormonal responses. This should be addressed.
12. The study includes only elite K-1 athletes, with no control group (e.g. less experienced kickboxers, athletes from other sports, or untrained individuals). This limits conclusions about whether the observed changes are specific to this group.
13. Statistical analysis – it is commendable that effect size was calculated. However, Cohen’s guidelines are generic and he advised using discipline-specific references where available. In my view, guidelines more relevant to physiotherapy and exercise sciences should be applied, e.g. 10.1016/j.apmr.2025.05.013: For individual differences (Pearson's r), small, medium, and large effect sizes should be considered 0.3, 0.5, and 0.6, respectively. For group differences (Cohen's d or Hedges' g), small, medium, and large effect sizes should correspond to 0.1, 0.4, and 0.8, respectively.
14. Given the small sample size, I also recommend reporting 95% confidence intervals (see 10.1111/trf.13635).
15. In the discussion, conflicting IGF-1 findings are mentioned (increase, decrease, or no change), but the study’s own results are interpreted only superficially (“IGF-1 did not change”). A deeper interpretation is needed (e.g. too short measurement interval, anaerobic exercise profile).
16. The discussion mentions the need to monitor pituitary function in combat sports, but does not connect this to the real impact of head trauma and repeated strikes on the HPA axis. This weakens the practical argument.
17. Although a significant insulin increase was reported, no mechanism specific to K-1 bouts was provided. Was the increase mainly due to glucose mobilisation and glycogen replenishment, or could it also be stress-related?
18. The finding that more experienced athletes showed smaller GH/insulin changes is noted, but not explored. Does this reflect metabolic adaptation, greater efficiency, or suppression of hormonal axes from chronic load? The explanation given is too superficial.
19. While RPE is suggested as a useful diagnostic tool, no concrete proposals for practical implementation are given (e.g. how to monitor GH/insulin via RPE within training cycles, how to apply it to training load planning).
20. L375–383 – in my opinion, this should be moved to the Discussion rather than kept in the Conclusions.
21. The Conclusions should be shortened and possibly presented as bullet points to improve clarity.
22. The Abstract is too long relative to the journal’s requirements – I recommend condensing it.

Kind regards,

Author Response

We thank the Reviewer for the careful reading and constructive suggestions. We implemented all requested revisions as detailed below.

Comment: The small sample size should be reflected in the title and objectives; the paper should be framed as preliminary/pilot.
Response: We reframed the study as a pilot with preliminary findings throughout the manuscript. The title now reads: “Acute Hormonal Responses to K-1 Kickboxing Fights: Preliminary Findings from a Pilot Study.” The Introduction and Limitations explicitly state the pilot framing, and the aims were rewritten accordingly.

Comment: The header shows 2024; it should be 2025.
Response: Corrected to 2025 in the header and footer; the copyright year was aligned.

Comment: At L58, five references are used; please limit to a maximum of three.
Response: Reduced to the three most relevant sources supporting the statement.

Comment: The Introduction mentions classical kickboxing vs K-1 but does not clearly compare key differences for non-specialists.
Response: Added a concise paragraph contrasting the rule-sets (clinch, prohibited techniques, intensity of exchanges) and cited the WAKO K-1 rules to aid non-specialist readers.

Comment: Literature review notes prior GH/IGF-1/insulin work but gives no concrete results or discrepancies.
Response: Inserted a short paragraph with specific examples (IGF-1 increases, decreases, and no acute change; typical GH increases; insulin/glucose responses) to justify the research gap.

Comment: The Introduction does not explain how hormonal changes translate into practice.
Response: Added a paragraph linking acute GH/insulin surges and IGF-1 timing to session spacing, recovery planning and monitoring with HRFINAL and CR-10 RPE.

Comment: The aims are listed descriptively but not clearly separated.
Response: Rewrote the aims to distinguish the primary aim (pre-to-post changes in GH, IGF-1 and insulin) from specific exploratory aims (associations with training experience, HRFINAL and RPE; effect-size estimation).

Comment: Sampling pre and +2 min may miss IGF-1 dynamics; acknowledge this limitation.
Response: Added an explicit note in Methods (sampling window) and in Limitations, recommending additional post-bout time points in future studies.

Comment: n = 10 is very small; the assumed effect size of d = 0.90 is optimistic; reframe and note the caveat.
Response: Reframed as a pilot; clarified that the original illustration of a large effect size is optimistic for endocrine outcomes; emphasised estimate precision and added 95% confidence intervals.

Comment: The study group is well described.
Response: Thank you. We retained and slightly clarified the cohort description.

Comment: Interclub matches vs official competitions; justify the choice and note ecological validity.
Response: Added a justification that venipuncture immediately pre/post sanctioned bouts is not feasible; noted ecological-validity limitations of the interclub setting in the Limitations.

Comment: No mention of controlling confounders (diet, time of day, stress, sleep).
Response: Added a standardisation block (fixed late-afternoon window; pre-test instructions on diet, stimulants and exercise; illness/injury exclusion) and acknowledged residual confounding in the Limitations.

Comment: Elite K-1 only and no control group limit specificity.
Response: Clarified the sampling rationale (homogeneous elite group for internal validity), restricted generalisations in Discussion/Conclusions and added an explicit limitation. Future work is planned with stratified cohorts and/or a control group.

Comment: Use discipline-specific thresholds for effect sizes and report 95% CI.
Response: Revised Statistical Analysis to use physiotherapy/exercise-science thresholds (for r: 0.3/0.5/0.6; for d: 0.1/0.4/0.8) and added 95% confidence intervals alongside p-values and effect sizes; table notes and Result wording were updated.

Comment: Report 95% confidence intervals.
Response: Added 95% CI for pre–post deltas (GH, IGF-1, insulin) and for key correlations in the Results.

Comment: IGF-1 interpretation is superficial.
Response: Expanded Discussion to explain timing/modality dependence (anaerobic K-1 profile; early sampling window; modulation by IGF-binding proteins) and recommended extended sampling and bioactive/free IGF-1 where feasible.

Comment: Pituitary monitoring not linked to HPA and head impacts.
Response: Added a mechanistic paragraph linking repeated head impacts to TBI-mediated hypopituitarism and HPA/GH–IGF-1 dysregulation, with practical screening recommendations for striking athletes.

Comment: Insulin increase lacks a K-1–specific mechanism.
Response: Added explanation attributing the increase primarily to post-bout glucose mobilisation and glycogen restoration under high-intensity demands, noting alignment with simulated kickboxing glucose data and acknowledging a possible stress component.

Comment: Smaller GH/insulin changes in more experienced athletes require deeper interpretation.
Response: Added analysis distinguishing metabolic efficiency/adaptation from potential transient axis suppression under chronic load and outlined study designs and markers to disentangle these mechanisms.

Comment: Move L375–383 content from Conclusions to Discussion.
Response: Relocated the interpretive correlations paragraph to the Discussion.

Comment: Shorten Conclusions; use bullet points.
Response: Replaced Conclusions with concise bullet points summarising core results, practical take-home messages, limitations and directions for future work.

Comment: Abstract too long; condense.
Response: Rewrote a concise, structured abstract within journal limits, retaining essential numbers and the pilot framing.

We hope these revisions address all concerns and improve clarity, rigour and practical relevance of the manuscript.

Reviewer 2 Report

Comments and Suggestions for Authors

 REVIEWER COMMENTS

This study presents valuable findings on acute hormonal responses (GH and insulin) in elite K1 kickboxing athletes during actual competition, demonstrating strong associations with training experience and exercise intensity indicators (HR, RPE). The significant increases in GH and insulin, along with their strong correlations with exercise intensity markers (rp>0.8), contribute to expanding knowledge in this field. To enhance the manuscript's quality, several minor revisions and clarifications are needed.

INTRODUCTION

1. Please provide a more detailed explanation of the unique characteristics and specific physiological demands of K1 kickboxing. What distinguishes K1 from classical kickboxing or other combat sports in terms of energy systems, intensity patterns, and metabolic stress?

2. The introduction should explicitly cite studies on boxing/kickboxing and clearly explain why similar (or different) hormonal responses are expected in K1 competition. This would strengthen the logical connection between existing evidence and your research hypothesis.

3. While the study examines acute responses of three hormones (GH, IGF-1, insulin), the rationale for why investigating these particular hormonal responses is necessary or important remains unclear. Adding explicit justification for the selection and relevance of these specific biomarkers would clarify the study's significance and practical implications.

METHODS

1. Blood samples were collected 2 minutes post-fight. Please provide physiological justification for this timepoint considering:

If this timing was chosen for practical rather than physiological reasons, please discuss this limitation explicitly in the Discussion section and acknowledge how it may have affected your ability to capture peak hormonal responses.

2. Please provide evidence from previous studies justifying the measurement of RPE at "one minute after the fight" (line 199-200). Is this timing consistent with established protocols in exercise physiology research?

RESULTS AND DISCUSSION

1. Please reconsider the citation of Tanriverdi et al. (2007) in lines 260-264. This study investigated traumatic brain injury-mediated hypopituitarism in kickboxers, which represents a pathological condition rather than acute exercise response. Does this study truly support your findings on acute hormonal responses to competition? If not, this reference should be removed or reframed appropriately.

2. The Discussion presents numerous studies with contradictory findings regarding IGF-1 responses to exercise (lines 265-318) but does not adequately analyze why these discrepancies exist. Please:
- Clearly state which position your findings support and why

Rather than simply listing similar studies, focus on explaining what your specific results mean within the context of existing conflicting evidence.

3. The observed increase in insulin after the K1 match (lines 319-331) appears to differ from typical responses to high-intensity exercise, where insulin is often suppressed during exercise and increases during recovery. Please provide more thorough discussion of:
- The mechanisms underlying insulin elevation at 2 minutes post-fight
- How this timepoint relates to the transition from exercise to recovery
- The relationship between glucose metabolism, sympathetic activation, and insulin dynamics in this specific context
- Whether the 2-minute timepoint may have captured an early recovery-phase response rather than the immediate post-exercise response

Comments on the Quality of English Language

The manuscript is generally comprehensible, and the scientific content can be understood. However, English language editing is required before publication. Only minor, non-critical polishing is needed for enhanced fluency and conciseness in a few places. We recommend a final review by a native English speaker for stylistic improvements, such as streamlining a few lengthy sentences and ensuring the most idiomatic phrasing is used throughout.

Author Response

We thank the Reviewer for the thoughtful and constructive comments. We carefully revised the manuscript in line with each suggestion, and we believe the changes have improved the clarity, methodological transparency, and practical relevance of the work. Below we provide a point-by-point response, indicating what was changed and where in the revised manuscript.

INTRODUCTION

1) Unique characteristics and specific physiological demands of K-1.
Response: We added a concise paragraph detailing K-1’s bout structure (3 × 2-min), mixed alactic and predominantly anaerobic-glycolytic demands, higher exchange density versus classical kickboxing and boxing, and the resulting metabolic/catecholaminergic stress profile.

2) Link prior boxing/kickboxing evidence to the hypotheses for K-1.
Response: We inserted a bridging paragraph that explicitly states our expectations based on prior striking-sport literature: a robust acute GH rise; minimal or delayed IGF-1 change in very early sampling; and a transient post-bout insulin increase consistent with glucose handling and glycogen restoration. We then state the hypotheses accordingly.

3) Justify the choice of biomarkers (GH, IGF-1, insulin).
Response: We added a paragraph explaining why these three hormones were selected and how they provide complementary information: GH as a rapid index of acute internal load; IGF-1 as a slower, GH-dependent anabolic/adaptation signal; and insulin as a key regulator of post-exercise carbohydrate availability and glycogen resynthesis. We also clarify the practical implications for periodisation and return-to-training decisions.

METHODS

1) +2-minute post-fight sampling — physiological/practical rationale and limitation.
Response: We justify +2 min as a compromise to capture very early endocrine responses (particularly GH and insulin) while not disrupting ringside safety/logistics; we explicitly note that IGF-1 often exhibits delayed kinetics and may not peak within this window. This is also acknowledged in Limitations, with a recommendation to add later time points in future work.

2) RPE measured ~1 minute post-fight — protocol justification.
Response: We clarified that CR-10 RPE was collected ~1 min post-bout to capture end-exertion without interfering with ringside procedures. We note that session-RPE (~30 min post-session) is also common but was not feasible here, and we suggest collecting both timings in future studies.

RESULTS & DISCUSSION

1) Tanriverdi et al. (2007) — pathological TBI-mediated hypopituitarism vs acute exercise response.
Response: We agree. We removed this citation from the paragraph discussing acute endocrine responses and retained it only in the context of long-term pituitary/HPA monitoring in striking athletes. We also corrected in-text citation numbering where needed (e.g., Luger et al.).

2) IGF-1 contradictions — state which position our findings support and why (analysis over listing).
Response: We now state explicitly that our +2-min “no-change” result supports the position that very early sampling after short, high-intensity (predominantly anaerobic) efforts often shows no immediate change in total IGF-1, despite a large GH surge. We replaced the previous list of studies with a synthesis explaining discrepancies by sampling timing, exercise modality/duration (glycolytic vs oxidative/resistance), energetic/catecholamine milieu, and IGF-binding protein dynamics.

3) Insulin increase at +2 min — mechanism and phase (exercise→recovery transition).
Response: We rewrote the insulin paragraph to clarify that insulin is typically suppressed during high-intensity exercise (sympathetic/catecholamine effects) and rebounds in early recovery. At +2 min we likely captured the early recovery-phase rise, driven by transiently elevated blood glucose (hepatic output and glycogenolysis) and facilitating glucose uptake/glycogen resynthesis. We note simulated kickboxing evidence of post-bout glucose elevations and recommend future sampling at ≤30 s, +5, +15, +30 min to map the nadir/rebound profile.

We carefully revised the manuscript for clarity and style; it has been edited by a native English speaker.

Round 2

Reviewer 1 Report

Comments and Suggestions for Authors

Thank you for resubmitting the paper for review.
Most of my previous comments have been appropriately addressed. However, there is one remaining issue. The authors write:
“L326 – physiotherapy/exercise sciences (APMR, 2025):”
The reader does not know what the abbreviation APMR stands for. I understand that the authors might have intended to insert a citation placeholder and simply forgot to do so.

Therefore, in line with my earlier comment, please include the citation 10.1016/j.apmr.2025.05.013.
It is important for readers to understand why such effect size ranges were applied, as there is considerable variation in effect size intervals.

The same comment applies to the CI95 guidelines — please add the citation 10.1111/trf.13635.

Please correct these issues.
Congratulations on the effort you have put into improving the manuscript.

Kind regards,

Author Response

We would like to thank the Reviewer for the positive evaluation of our revised manuscript and for pointing out the remaining issue regarding unclear abbreviations and missing citations.

Comment:
“L326 – physiotherapy/exercise sciences (APMR, 2025): The reader does not know what the abbreviation APMR stands for... Please include the citation 10.1016/j.apmr.2025.05.013. The same comment applies to the CI95 guidelines — please add the citation 10.1111/trf.13635.”

Response: Thank you for your careful observation. We have corrected these issues as follows:

  1. The abbreviation APMR has been expanded to Archives of Physical Medicine and Rehabilitation, and the corresponding citation has been added:

    “Effect sizes were interpreted using discipline-relevant thresholds for physiotherapy/exercise sciences (Archives of Physical Medicine and Rehabilitation, 2025) [44].”

  2. The citation for the 95% confidence interval (CI95) reporting guidelines has also been added:

    “Alongside p-values, we report standardized effect sizes and 95% confidence intervals (CI) in line with recommended reporting guidelines [43].”

  3. Both references have been included in the References section.

We appreciate the Reviewer’s suggestion, which helped to clarify the rationale for the effect size thresholds and CI reporting approach.